# Physiological responses to ultra-high CO$_2$ levels in an evergreen tree species

Ben-El Levy[1], Yedidya Ben-Eliyahu[1], Yaniv-Brian Grunstein[1], Itay Halevy[2], Tamir Klein[1]

[1] Department of Plant and Environmental Sciences, Weizmann Institute of Science, 76100 Rehovot, Israel

[2] Department of Earth and Planetary Sciences, Weizmann Institute of Science, 76100 Rehovot, Israel

**Corresponding author**: Tamir Klein, Department of Plant and Environmental Sciences, Weizmann Institute of Science, 76100 Rehovot, Israel. ORCID: 0000-0002-3882-8845. Tamir.klein@weizmann.ac.il

**Abstract**

Although numerous experiments have been dedicated to studying plant response to elevated CO$_2$, almost none crossed the level of 1000 ppm. Plant responses to high CO$_2$ levels importantly inform our understanding of plant physiology in ultra-high CO$_2$ environments, e.g., in Earth history, in the case of unmitigated anthropogenic emissions, and for future colonization of Mars.

Here, we challenged two-year old seedlings of fruit trees grown in soil in a mesocosm, with CO$_2$ levels of 400, 1600 and 6000 ppm, the highest of which is approximately equivalent to that of Mars' atmosphere. Plant growth, and leaf gas exchange (transpiration, stomatal conductance, and CO$_2$ assimilation) were measured on a weekly basis for 23-25 consecutive days. We hypothesized that elevated CO$_2$ levels will induce a decrease in transpiration, primarily attributed to reduced stomatal conductance. Indeed, leaf transpiration was decreased at 1600 ppm CO$_2$ and remained low at 6000 ppm,

concurrent with a 50% decrease in stomatal conductance. The CO$_2$-induced stomatal closure appears to have saturated between 850 and 1600 ppm CO$_2$. Due to this effect, net assimilation was only mildly changed at 1600 ppm CO$_2$, but significantly increased at 6000 ppm. As a result, water-use efficiency quadrupled at 6000 ppm CO$_2$. Stem height increment did not change significantly across the CO$_2$ treatments.

Taken together, our measurements demonstrated both the potential and limit of CO$_2$-induced stomatal closure, with positive

implications for fruit tree growth in ultra-high CO$_2$ environments, as on Earth in the case of unmitigated anthropogenic CO$_2$ emissions and on Mars.

**Plain language summary**

As atmospheric CO$_2$ increases globally, plants increase the rate of photosynthesis. Still, leaf gas exchange can be downregulated by the plant. Here we tested the limits of these plant responses in a fruit tree species under very high CO$_2$

levels relevant to future Earth and to contemporary Mars. Plant water use decreased at 1600 ppm CO$_2$ and remained low at 6000 ppm. Photosynthesis significantly increased at 6000 ppm. In summary, ultra-high CO$_2$ may partly compensate for limited water availability.

**Introduction**

Understanding the implications of elevated atmospheric $CO_2$ levels on plant physiology is paramount, as it has profound consequences for global ecosystems, water economy, and terrestrial carbon cycling (Bazzaz et al. 1990). At the microscopic level, the sensitivity of plants to altered $CO_2$ levels directly impacts carbon assimilation through photosynthesis. When the level of $CO_2$ within the plant diminishes, a regulatory response is triggered within the stomatal guard cells, and the stomata open, allowing diffusive uptake of atmospheric $CO_2$ (Lawson and Morison 2004). This gas exchange comes at a cost. When the stomata are open, the plant loses water to the air in transpiration (Brodribb et al. 2009). This loss of water, while essential for photosynthesis and temperature regulation, can become a significant challenge for the plant in water-limited environments (Wagner et al. 2021, 2022).

The concentration of $CO_2$ in the leaf and the surrounding atmosphere is a critical factor for photosynthetic carbon fixation. Changes in $CO_2$ levels can trigger adjustments in stomatal conductance to optimize photosynthesis and water-use efficiency (Klein et al. 2013). Yet, stomata are sensitive to many additional factors. Stomata generally open in the presence of light, as photosynthesis, the process that converts $CO_2$ and light energy into sugars, is active during daylight hours. Moreover, the relative humidity inside and outside the leaf affects stomatal conductance. Lower external humidity levels can downregulate stomatal conductance, leading to a decrease in water loss through transpiration (Wagner et al. 2022). Leaf temperature can also directly impact stomatal conductance. Warmer conditions often lead to increased transpiration rates, causing stomata to open wider to cool the leaf (Uni et al. 2023). Fluctuations in nutrient availability, particularly potassium, can influence stomatal conductance by affecting the turgor pressure of guard cells, which control stomatal opening and closing (Lebaudy et al. 2008). Often, these responses act together in shaping stomatal conductance at a given atmospheric $CO_2$ concentration (Bartlett et al. 2016).

On a macroscopic scale, elevated $CO_2$ exerts significant influence over planetary climate, presenting both challenges and opportunities for life on Earth and beyond (Forget et al. 2013, Ozak et al. 2016). For example, on Mars, where the partial pressure of $CO_2$ is over an order of magnitude higher than that of Earth's atmosphere, understanding the impact of high $CO_2$ on plant physiology and growth may aid efforts of human colonization, for which suitable conditions are thought to exist (Slobodian et al. 2015). Mars' atmosphere is about 95% $CO_2$, and at an average surface pressure of ≈6.5 mbar, this is equivalent to ≈6200 ppm $CO_2$ on Earth (Franz et al. 2017). At the same time, Mars' atmosphere contains only trace amounts of $O_2$ (0.16%) and $H_2O$ (0.03% on average, but highly variable). Water is not only rare in the Martian atmosphere, but also on its surface, present mainly as ice or in hydrated salts, inaccessible to plants (Diez 2018, Nazari-Sharabian 2020). Despite the growing interest in colonizing Mars, plant growth under such conditions has been explored in only a few studies (e.g., Richards et al. 2006, Wamelink et al. 2014).

Ultra-high atmospheric $CO_2$ levels are also relevant on Earth, past and future. Looking ahead, if industrial development continues as usual (IPCC emission scenario RCP8.5), the $CO_2$ concentration may rise to 1000 ppm by the end of this century and double that by the end of the year 2200. The effects of these changes on plant development are unknown, and so are the implications for the carbon and water cycles on Earth. This said, such high $CO_2$ levels existed in Earth's deep past. Recent
reconstructions indicate that between ≈65 and ≈40 million years ago atmospheric $CO_2$ levels were close to ≈1000 ppm, peaking at ≈1600 ppm around 51 million years ago (CenCO2PIP et al. 2023). Over this time interval, all extant plant families already existed (Li et al. 2019). Therefore, we can assume that extant plant species have evolved through large changes in atmospheric $CO_2$ levels, up to four-fold the current level or higher. At a $CO_2$ level of 1000 ppm, net assimilation is close to saturation, with values typically at 79-91% of the maximum rate, depending on temperature (Kirschbaum 1994).
However, at 400 ppm and 25 ºC, net assimilation is at 64% of the maximum rate.

Interestingly, the high abundance of $CO_2$ in the Martian atmosphere (comparable to ≈6200 ppm on Earth) may compensate for water shortage on the planet's surface. Plants open their stomata to capture $CO_2$ from the atmosphere, inevitably losing water vapor (Brodribb et al. 2009). Under rising $CO_2$ levels, stomatal opening can be minimized, thus saving water for the
plant and providing an advantage in dry conditions (Elliott-Kingston et al. 2016, Paudel et al. 2018). Preliminary research hints at the potential outcomes of our exploration: short-term (minutes) responses to an increase in atmospheric $CO_2$ levels, often involve a decrease in stomatal conductance (Medlyn et al. 2001, Paudel et al. 2018). This initial reduction is a reversible response and is linked to an increase in carbon fixation during photosynthesis. Essentially, when more $CO_2$ is available, the plant can achieve the same level of carbon uptake at lower stomatal opening, thereby conserving water in the
process. In addition, leaf surface pH might decrease due to increased concentrations of dissolved inorganic carbon in leaf surface solutions (Zeebe and Wolf-Gladrow 2001). However, leaves can regulate their leaf surface pH, through mechanisms that are still under study (Gilbert and Renner 2021). On longer timescales, from weeks through the lifetime of an individual plant (Hasanuzzaman et al. 2023) to generations over evolutionary timescales (Steinthorsdottir et al. 2019), plants can undergo morphological and developmental changes in response to altered $CO_2$ concentrations. Fossilized leaves from times
in Earth history when $CO_2$ levels were higher display a lower areal density of stomata (Beerling et al. 1993, Haworth et al. 2012). As atmospheric $CO_2$ concentrations decreased over time, many plant species evolved to increase stomatal density. It is believed that this long-term adaptation to slowly evolving $CO_2$ levels enabled plants to thrive under changing environmental circumstances. Stomatal density has been hence used extensively as a paleo-$CO_2$ proxy (Steinthorsdottir et al. 2019, Konrad et al. 2021). However, this use has been rarely validated experimentally, but rather calibrated against other
paleo-$CO_2$ proxies (Konrad et al. 2021; but see Brownlee 2001).

Despite the decrease in leaf gas exchange, on a timescale of weeks, carbon assimilation still increases under elevated $CO_2$, due to passive diffusion, as shown in both fruit trees (Paudel et al. 2018) and forest trees (Dror and Klein 2022) grown under controlled conditions. Among plant species, trees demonstrate the highest $CO_2$-induced photosynthetic increase, about 45%

increase from 370 to 570 ppm $CO_2$, as shown in a meta-analysis of free-air $CO_2$ enrichment experiments (Ainsworth and Rogers 2007). The higher assimilation can lead to increased carbon storage (Kinsman et al. 1997, Paudel et al. 2018) and increased growth. For example, elevated $CO_2$ levels resulted in elongation of branches and stems by 33% in *Garcinia mangostana* (Downton et al. 1990) and by 15% in *Glycine max* (Rogers et al. 1992). However, in contrast to growth in controlled environments, it has been established that under field conditions, where competition over limited resources prevails, tree growth is mostly unaffected by elevated $CO_2$ (Korner et al. 2005, Klein et al. 2016, Jiang et al. 2020, Norby et al. 2022). Yet, increased tree growth is still observed in some cases (Kim et al. 2020, Norby et al. 2024). Common to all these experiments is the moderate level of elevated $CO_2$, up to 1000 ppm, motivated by the gradual increase of atmospheric $CO_2$ concentrations on Earth (Klein and Ramon 2019). An exception, a few studies were conducted at ultra-high $CO_2$ levels of up to 50,000 ppm. Experimenting with seedlings in small chambers, grown on artificial media, it was shown that the fresh weight of lettuce, mint, and thyme increased up to 10000 ppm $CO_2$, decreasing somewhat at 30000 ppm, but still five-fold higher than at ambient $CO_2$ (Tisserat and Silman 2000, Tisserat and Vaughn 2001). A similar trend was observed in young seedlings of loblolly pine grown on artificial media (Tisserat and Vaughn 2003). The effect was smaller in plants grown in soil, with a two-fold increase in fresh weight at 3000 ppm $CO_2$ (Tisserat and Vaughn 2001). These experiments importantly constrained aspects of plant growth at ultra-high $CO_2$ levels, but they did not address the matter of the plant's water budget.

High concentrations of $CO_2$, which increase $CO_2$ assimilation and decrease the stomatal conductance, lead to a notable reduction in water loss and an increase in the plant's water-use efficiency (WUE), the ratio of net $CO_2$ assimilation to transpiration, representing the plant's ability to maximize carbon gain while minimizing water loss (Field et al. 1995, Mathias and Thomas 2021, Dror and Klein 2022). Experiments have demonstrated that tree seedlings exposed to double the ambient $CO_2$ levels exhibit a decrease in stomatal conductance ranging from 12% to 36% (Klein and Ramon 2019). This reduction in stomatal conductance signifies that at elevated $CO_2$ levels, fewer signals are sent to keep stomata open, resulting in reduced plant water loss. However, it is crucial to recognize that the response to elevated $CO_2$ is not uniform across all plant species. Factors such as the duration of exposure to high $CO_2$, the availability of other essential resources like water and nutrients, and inherent genetic differences between species can modulate the plant's response. Some plant species may even exhibit reduced growth or alterations in their architecture, including shorter stems or smaller leaves, in response to elevated $CO_2$ levels.

Among tree species, evergreen broadleaf species of tropical biomes have shown higher sensitivity to $CO_2$ levels in terms of reduced stomatal conductance (Klein and Ramon 2019). Taking advantage of this sensitivity, in this study, an experiment was designed to investigate the response of dwarf guava (*Psidium cattleyanum*), an evergreen broadleaf of the Amazon rainforest, to ultra-high $CO_2$ levels. This short-stature tree species (growing up to 6 m in height) is also well-adapted to medium light levels. Dwarf guava is an economically important species in some areas where it is cultivated (Patel 2012) and an invasive species in others, like Hawaii (Huenneke et al. 1990). We used an isolated mesocosm chamber that was uniquely

constructed to maintain $CO_2$ carbon dioxide as the only variable while the other variables (temperature, relative humidity, illumination intensity, nutrient supply) remained constant. In this mesocosm chamber we placed ten two-year-old guava seedlings and increased the $CO_2$ levels to about four times the ambient atmospheric concentration, 1600 ppm, and to a Mars-like atmospheric concentration, 6000 ppm (although not attempting to simulate Mars' conditions *per se*). We hypothesized that elevated $CO_2$ concentrations would induce a decrease in leaf transpiration, primarily by a reduction in stomatal conductance. This research may advance our understanding of plant growth and water-use efficiency under conditions expected on Earth by the year ≈2100 if anthropogenic $CO_2$ emissions are unmitigated, and beyond the terrestrial realm, it serves as a steppingstone towards ecological solutions in planetary environments.

**Materials and methods**

*Experimental Design*

Ten two-year-old dwarf guava plants (*Psidium cattleyanum*) of similar size were transferred to 1-liter pots. Plants were placed in a sealed mesocosm chamber ($0.90 \times 0.65 \times 0.60$ m = $0.351$ m$^3$; COY, MI, USA; Rog et al. 2020). This chamber is designed to control humidity and temperature conditions. The humidity was maintained at 70% and the temperature at 24-26 ˚C, producing a vapor pressure deficit of 1.3-1.4 kPa. Tubes from $CO_2$ and $N_2$ cylinders were connected to the chamber to regulate gas concentrations, allowing for three experimental treatments with varying $CO_2$ levels: ambient, high, and ultra-high (400, 1600, and 6000 ppm $CO_2$, respectively). Experimenting with such high $CO_2$ levels required special safety precautions, and a $CO_2$ detector alarming of leaks was installed outside the chamber. This was complemented by periodic $CO_2$ measurements around the chamber, constantly showing levels of 400-480 ppm, typical of an indoors environment and assuring no leak from the chamber. To eliminate potential effects of differences among plants and chambers, the entire experiment was conducted on the same plants within the same chamber. Therefore, treatment periods were limited to just over three weeks (23-25 days), so that all treatments were applied on plants at the same developmental stage. This period is sufficient to allow for plant acclimation to the $CO_2$ treatment (Poorter et al. 2022). The experiment was conducted between 4 June and 27 August 2023. Plants were first exposed to 400 ppm $CO_2$, followed by 1600 ppm, and then 6000 ppm, i.e., from ambient to high and ultra-high $CO_2$. Prior to each treatment, plants underwent an additional one-week acclimation period within the mesocosm chamber. Chamber $CO_2$ was regulated using a SCD30 sensor with measurement accuracy of ±30 ppm at a range of 400-10000 ppm (Sensirion, Chicago, IL, USA) and a ESP32 controller (Espressif Systems, Shanghai, China). Notably, at 400, 1600, and 6000 ppm $CO_2$, fluctuations of ±30 ppm are equal to ±7.5%, 1.9%, and 0.5%, which are small. For this reason, we consider the $CO_2$ levels to have been steady. We utilized an HLG-320H-36A solar lamp (Agro Light, Beit Yehoshua, Israel) as the primary light source, positioned on top of the chamber to provide a photosynthetically active radiation (PAR) range of 550-700 µmol m$^{-2}$ s$^{-1}$ at the canopy height. However, most leaves grew below the canopy leaves, under lower PAR of 50-400 µmol m$^{-2}$ s$^{-1}$, which was still adequate for this tree species (see next paragraph). The day/night cycle was set to 12h/12h. A light distribution map within the mesocosm chamber showed that plants were exposed to a

homogeneous light regime. Individual plant positions were rotated every other day to mitigate any positional effects within the chamber.

*Plant response to light intensity and instantaneous changes in CO₂ level*

To validate that *Psidium cattleyanum* is suitable for growth in the mesocosm chamber conditions, a photosynthetic light response curve was constructed in a pilot experiment. Six plants were exposed to increasing light levels and their net assimilation rate ($\mu mol\ CO_2\ m^{-2}\ s^{-1}$) was measured. Determination of assimilation rate was achieved with a LI-6800 infrared

gas analyzer (IRGA; Li-Cor Biosciences, Lincoln, NE, USA). Temperature, relative humidity, and $CO_2$ level within the leaf cuvette were adjusted to ambient during the measurement. PAR was adjusted to 1000, 800, 600, 400, 300, 200, 100, 50, and 0 $\mu mol\ m^{-2}\ s^{-1}$, in this order. The value of assimilation rate increased linearly from –1.89 at darkness (PAR=0) to 5.93 $\mu mol$ $CO_2\ m^{-2}\ s^{-1}$ at PAR of 200 $\mu mol\ m^{-2}\ s^{-1}$ (Fig. S1). The compensation point was at 45 $\mu mol\ m^{-2}$ $s^{-1}$, increases in assimilation rate were minor, and assimilation rate saturated. A second pilot experiment tested the plant

response to instantaneous changes in $CO_2$ level. Here, three plants grown at ambient $CO_2$ level were exposed to increasing $CO_2$ levels and their net assimilation rate ($\mu mol\ CO_2\ m^{-2}\ s^{-1}$) and stomatal conductance ($mol\ m^{-2}\ s^{-1}$) were measured with the LI-6800 infrared gas analyzer. Temperature, relative humidity, and PAR level within the leaf cuvette were set to 26 ºC, 75%, and 600 $\mu mol\ m^{-2}\ s^{-1}$, respectively. $CO_2$ level was adjusted every 3-4 minutes to 400, 100, 50, 0, 150, 250, 300, 400, 500, 600, 700, 800, 900, 1000, 1100, 1200, 1300, 1400, 1500, 1600, 1700, 1800, 1900, 2000 and 400 $\mu mol\ CO_2\ m^{-2}\ s^{-1}$, in this

order.

*Plant growth*

Plants were actively growing throughout the experiment. We measured 4 growth parameters on a weekly basis: plant height (cm), leaf surface area ($cm^2$), branch number and shoot number. Stem diameter growth was too small for a manual

measurement, as expected in saplings of this size. The height of each guava plant was measured from the base of the stem to the tip of the plant using a tape measure. The number of new branches was counted manually, with the appearance of clear protrusions consisting of four or more leaves considered as new branches. The number of shoots extending from the ground was recorded separately. The leaf surface area was determined by counting the number of leaves on each plant and multiplying by an average leaf size. The average leaf size was obtained using the Easy Leaf Area application developed by

Hsien Ming Easlon (https://github.com/heaslon/Easy-Leaf-Area). In this process, leaf tops were photographed close to a 2×2 $cm^2$ area painted red over a white background. The leaf area was then derived from the ratio of green to red pixels present in the image.

*Leaf gas exchange*

Stomatal conductance and transpiration were measured using a LI-600 infrared gas analyzer (Li-Cor Biosciences, Lincoln, NE, USA). The LI-600 was applied to determine the gas exchange during photosynthesis in fresh leaves, with a known leaf

surface area of 1 cm$^2$. The rates of transpiration (E in mmol m$^{-2}$ s$^{-1}$) and stomatal conductance (gs in mmol m$^{-2}$ s$^{-1}$) were measured between 9:00 a.m. and 11:00 a.m. The rate of $CO_2$ assimilation was measured with a leaf cuvette at 400 ppm μmol $CO_2$ using the LI-6800 infrared gas analyzer (see under *Plant response to light intensity*) but could not be measured using the leaf cuvette at the higher $CO_2$ levels, due to the instrument limitations. Instead, assimilation rate was measured across the three $CO_2$ levels by analyzing the rate of decrease in the $CO_2$ concentration (ppm) within the controlled mesocosm chamber over a known duration (measured in s), which was then divided by the corresponding leaf surface area of all plants (m$^2$). The amount of assimilated $CO_2$ was transformed from the change in ppm $CO_2$ into μmol $CO_2$ by multiplying the ppm value by 14.558, which is the number of moles of air in the chamber, according to the ideal gas law, considering atmospheric pressure and mesocosm volume and temperature (above). The SCD30 sensor data, responsible for monitoring the $CO_2$ levels within the chamber, was collected at regular intervals of 1.5-2.3 minutes. Notably, a decrease in the $CO_2$ level was considered when the sensor data showed a negative difference persisting for 3 or more consecutive intervals, and the net change in the $CO_2$ level was computed and subsequently divided by the total duration of the decrease. Data collection for this process was conducted between 8:00 a.m. and 4:00 p.m. exclusively, during daylight hours and on days with no disturbances within the mesocosm chamber. For each $CO_2$ level, assimilation rate was calculated for 20 periods, from which the mean and standard error were calculated. These measurements facilitated a comprehensive evaluation of the plants' $CO_2$ assimilation. Water-use efficiency (WUE) was calculated as the ratio of the rate of carbon assimilation to the rate of transpiration (E) in the guava plants on four different days for each $CO_2$ concentration.

*Leaf parameters*

Leaf stomatal density was determined on newly matured leaves formed in the new $CO_2$ environment (3 days old leaves from the top of the canopy, grown at PAR of 550 μmol m$^{-2}$ s$^{-1}$; *n* = 10 plants) by calculating the number of stomata per unit area (mm²). The lower epidermal layer was carefully affixed to a slide using a transparent contact adhesive. Subsequently, the slide was captured under a light microscope (Leica DM500, Wetzlar, Germany) at a 20× magnification. Utilizing the ImageJ software, images were analyzed, and the number of stomata was divided by the defined area obtained from the slide, with a 0.005 mm line reference. The pH level on leaf surfaces was measured non-invasively using a Hanna device (Hanna Instruments Inc. Woonsocket, USA). The leaves were carefully positioned on a stable cardboard surface, and the device was affixed directly to them.

*Statistical analysis*

We tested for significant differences among $CO_2$ treatments for specific parameters. Prior to statistical tests, the distribution of data was presented as a histogram for each parameter. Then, a normal distribution was fit to each histogram, and the goodness of fit was tested by a Shapiro-Wilk test. *P*-values of the Shapiro-Wilk tests were >0.05 (often >0.2), indicating that all parameters were normally distributed. Differences in transpiration and stomatal conductance were analyzed using ANOVA at a significance level α = 0.05. For the analysis, $CO_2$ treatments were transformed into a nominal parameter

(ambient, elevated, and ultra-high). All analyses were performed in JMP software (SAS, Cary, NC, USA). Differences in stomatal density, A, WUE, height increment, and leaf surface pH were analyzed by t-tests on pairwise comparisons. *P*-values are reported in the figures.

**Results**

Examining leaf transpiration (Fig. 1a), a 20% decrease was observed upon the transition from 400 to 1600 ppm $CO_2$ (from ~4.4 to ~3.5 mmol $m^{-2}$ $s^{-1}$; F=44.9, $P<0.0001$). The subsequent increase in $CO_2$ concentration to 6000 ppm resulted in no statistically significant changes in leaf transpiration compared to the 1600 ppm treatment, suggesting a potential threshold

effect in the modulation of transpiration rates by elevated $CO_2$. In terms of stomatal conductance (gs), the transition from 400 to 1600 ppm $CO_2$ resulted in a 50% reduction in gs, which persisted at the 6000 ppm treatment (from ~0.85 to ~0.42 mol $m^{-2}$ $s^{-1}$; F=22.6, $P<0.0001$; Fig. 1b). Overall, both transpiration and gs fluctuated across measurement days; yet, only under 400 ppm $CO_2$ was there an increasing trend. Considering that this treatment was applied immediately after plants were brought in from the greenhouse, values during the first 10 days under 400 ppm $CO_2$ might still reflect acclimation to the chamber

conditions. In contrast to the decrease in gs with $CO_2$ increase, stomatal density increased mildly with $CO_2$, with a significant 20% increase between the 400 and 6000 ppm $CO_2$ treatments (Fig. 2).

The assimilation rate of $CO_2$ (A) was calculated for the entire mesocosm and showed a value of 1.5 µmol $CO_2$ $m^{-2}$ $s^{-1}$ at 400 to 1600 ppm $CO_2$. While this value is relatively low, it was relatively close to the photosynthetic rate of *Psidium cattleyanum*

leaves under PAR of ~100 µmol $m^{-2}$ $s^{-1}$, which is close to the light intensity experienced by most leaves in the chamber (Fig. S1 and Methods). The assimilation rate exhibited a mild, yet significant decrease over the transition from 400 to 1600 ppm $CO_2$, demonstrating a major increase only at 6000 ppm (Fig. 3a). Water-use efficiency was unchanged upon the shift from 400 to 1600 ppm $CO_2$ and displayed a four-fold increase upon the subsequent shift to 6000 ppm (Fig. 3b). Measurements of the weekly growth height did not reveal a significant change (Fig. 4), nor was the weekly increment of leaf surface area (Fig.

S2). The height increment was ~1.6 cm week$^{-1}$, which is equal to 4-5% growth, regardless of $CO_2$ level. Stalled branch growth or even branch degradation appeared to have occurred upon the change to 6000 ppm $CO_2$ (data not shown).

Plants that were grown at ambient $CO_2$ level were also tested for their A and gs responses to instantaneous changes in $CO_2$ level (Fig. S3). A increased linearly from $-2.3\pm0.6$ µmol $CO_2$ $m^{-2}$ $s^{-1}$ at 0 ppm $CO_2$ to $12.9\pm1.0$ µmol $CO_2$ $m^{-2}$ $s^{-1}$ at 800 ppm

$CO_2$ (with a compensation point around 100 ppm $CO_2$). At higher $CO_2$ levels, A nearly saturated, yet continued to increase up to $15.7\pm0.8$ µmol $CO_2$ $m^{-2}$ $s^{-1}$ at 2000 ppm $CO_2$, the highest level permitted by the IRGA instrument. In parallel, gs decreased in two measured plants from 0.14 to 0.11 mol $m^{-2}$ $s^{-1}$, and from 0.09 to 0.05 mol $m^{-2}$ $s^{-1}$.

## Discussion

Decades of research on physiological tree responses to elevated $CO_2$ have shaped our understanding of vegetation feedbacks to the increasing level of atmospheric $CO_2$ on Earth (Medlyn et al. 2001, Korner et al. 2005, Klein et al. 2016). However, since most experiments were limited to $CO_2 < 1000$ ppm, the limit of stomatal sensitivity to elevated $CO_2$ has remained unclear, and specifically, the limit of the effect of $CO_2$-induced decrease in stomatal conductance on tree transpiration is unknown. Previous experiments with ultra-high $CO_2$ levels indicated consistent benefits in small plants grown in small chambers, mostly on artificial media (Tisserat and Silman 2000, Tisserat and Vaughn 2001, 2003). Yet these former experiments focused mostly on plant growth, leaving the physiological mechanisms and plant water use unresolved. Here, we challenged seedlings of fruit trees grown in a mesocosm, with ultra-high $CO_2$ levels of 1600 and 6000 ppm. We hypothesized that elevated $CO_2$ levels will induce a decrease in plant water use, primarily by a reduction in stomatal conductance. Indeed, transpiration was decreased at 1600 ppm $CO_2$ and remained low at 6000 ppm (Fig. 1a), due to reduced stomatal conductance (Fig. 1b). The effects on assimilation, WUE, and growth were also measured (Figs. 3, 4). Taken together, our measurements demonstrate both the potential and limit of $CO_2$-induced stomatal closure, with positive implications for fruit tree growth in ultra-high $CO_2$ environments, as on Earth in the case of unmitigated anthropogenic $CO_2$ emissions (e.g., IPCC emission scenario RCP8.5) and on Mars (Franz et al. 2017; although not attempting to simulate Mars' conditions *per se*).

Experimenting with lemon tree saplings, we previously showed that $CO_2$-induced stomatal closure can decrease transpiration, while still increasing assimilation, at $CO_2$ levels as high as 850 ppm (Paudel et al. 2018). Here we showed that this effect was active upon an increase from 400 to 1600 ppm $CO_2$ but that no further change occurred upon a further increase to 6000 ppm. On the contrary: stomatal conductance was sometimes higher at 6000 than at 1600 ppm $CO_2$ (but still lower than at 400 ppm; Fig. 1b). Across the measurements, stomatal conductance at 1600 ppm $CO_2$ was mostly ~50% its value at 400 ppm. The calculated change in stomatal conductance within this range (400-1600 ppm) was –33 mmol m$^{-2}$ s$^{-1}$ per 100 ppm $CO_2$ increase. For comparison, this $CO_2$-induced stomatal closure is higher than calculated for lemon (–19 mmol m$^{-2}$ s$^{-1}$ per 100 ppm $CO_2$ increase; Paudel et al. 2018), very similar to that measured in mango (–34 mmol m$^{-2}$ s$^{-1}$ per 100 ppm $CO_2$ increase) and identical with the mean calculated for broadleaf evergreen tree species across 39 different experiments (–33; Klein and Ramon 2019). However, due to the absence of a significant change in stomatal conductance from 1600 to 6000 ppm $CO_2$, the calculated change in stomatal conductance over the range 400-6000 ppm was only –5 mmol m$^{-2}$ s$^{-1}$ per 100 ppm $CO_2$ increase. Integrating our current and previous experiments, we can conclude that for an evergreen fruit tree such as guava, $CO_2$-induced stomatal closure probably saturates between 850 and 1600 ppm $CO_2$. Do these responses reflect an adaptation to elevated $CO_2$, or an instantaneous response? Experimenting with trees with no previous exposure to elevated $CO_2$, we observed a similar trend of stomatal closure (a 20-50% reduction in gs between 400 to 1600 ppm $CO_2$; Fig. S3) which seemed to level off at higher $CO_2$ levels. Despite this reduction, net carbon assimilation increased, mostly below 800 ppm $CO_2$, while measurements were limited to 2000 ppm $CO_2$, due to the IRGA constraints.

The measured stomatal response to the ultra-high $CO_2$ levels can explain the observed changes in carbon assimilation. Since assimilation of $CO_2$ into the leaf occurs by passive diffusion, most tree species increase assimilation despite $CO_2$-induced stomatal closure (Dror and Klein 2022). However, in our case the strong decrease in stomatal conductance at 1600 ppm was not compensated by the higher $CO_2$ availability, and the assimilation rate mildly decreased (Fig. 3a). Only at 6000 ppm, the surplus $CO_2$, which was not accompanied by additional stomatal closure, resulted in a significant increase in assimilation.

This indicates that the maximum rate of carboxylation increased under ultra-high $CO_2$ level, as previously shown in lemon under 650 and 850 ppm $CO_2$ (Paudel et al. 2018). The $CO_2$-induced photosynthetic increase observed here agrees with earlier studies at lower levels of elevated $CO_2$, which showed that photosynthesis is far from saturation at 400 ppm (Kirschbaum 1994), and that trees, more than shrubs, grasses, and crop plants, increase their photosynthesis under elevated $CO_2$ (Ainsworth and Rogers 2007). In turn, these physiological mechanisms yielded a major increase in WUE (Fig. 3b).

Noteworthy, the stable WUE (400-1600 ppm $CO_2$) was driven by proportional reductions in assimilation and transpiration, whereas the major WUE increase (at the 1600-6000 ppm $CO_2$ transition) was driven by increased assimilation. The stable WUE (400-1600 ppm $CO_2$) contrasts an overall increase of WUE in the terrestrial biosphere at this $CO_2$ range (Walker et al. 2021), however can be expected due to the reduced stomatal conductance in a broadleaf tropical tree species (Paudel et al. 2018), which is not a ubiquitous response across tree species (Klein and Ramon 2019). Finally, seedling growth, measured

here as the height increment, was consistent across the large $CO_2$ range (Fig. 4), meaning that the surplus carbon may have been allocated to sinks other than stem growth, such as storage (Paudel et al. 2018), respiration, or root growth (Dror and Klein 2022). In addition, there was an increase in new shoot production at 1600 ppm (data not shown). Our results are in apparent disagreement with those of the early ultra-high $CO_2$ experiments, in which the fresh weight of seedlings of several herb species and pine increased at ultra-high $CO_2$ and saturated only at levels of 10000 ppm (Tisserat and Silman 2000,

Tisserat and Vaughn 2001, 2003). Several differences between the early studies and our experiments may account for the different results. Unlike our experiments, in the earlier studies plants were grown on artificial media, sometimes supplemented with sucrose. The different growth conditions and substrate availability are expected to lead to differences in carbon assimilation of the observed sense (i.e., a greater biomass increase under the near-optimal conditions of the past studies). Indeed, the fresh weight increase measured in past experiments was smaller when plants were grown in soil, with

only a two-fold increase in fresh weight at 3000 ppm (Tisserat and Vaughn 2001). Lastly, at least for pine, stomata are almost insensitive to increases in $CO_2$ levels (Klein and Ramon 2019) and especially in loblolly pine (Will and Teskey 1997). Therefore, in pine variants, little or no $CO_2$-induced stomatal closure is expected and increases in carbon fixations are expected to be larger.

In contrast to the expected decrease in stomatal density in response to higher $CO_2$ (Beerling et al. 1993, Ainsworth and Rogers 2007), stomatal density increased modestly, but significantly, at the higher $CO_2$ levels in our experiment (Fig. 2b). For example, stomatal density increased by 15% between 400 and 1600 ppm $CO_2$, while stomatal size did not change. This

was unexpected, since overall, stomatal conductance decreased (e.g., by 50% between 400 and 1600 ppm $CO_2$). Therefore, it can be deduced that stomatal aperture was in fact reduced by 65%. Previous studies showed that stomatal conductance of mature leaves has a regulatory effect on the stomatal development of expanding leaves (Brownlee 2001, Miyazawa et al. 2005). We do not fully understand why stomatal density increased, and whether this change represents a long-term response. However, previous research shows that even for plants that were exposed for generations to elevated $CO_2$, responses can vary by species. Among 17 plant species growing in a naturally enriched $CO_2$ spring in Italy, stomatal density was higher under the higher $CO_2$ levels in 7 species, including two tree species (*Buxus sempervirens* and *Ruscus aculeatus*; Bettarini et al. 1998). Similarly, stomatal density increased in 10 out of 27 free-air $CO_2$ enrichment experiments (Ainsworth and Rogers 2007). More importantly, the assumption that plants decrease stomatal density in response to higher $CO_2$ has not been rigorously tested in biological experiments. Rather, stomatal density was tested against other paleo-$CO_2$ proxies such as contemporaneous $CO_2$ measurements from glacial ice cores, and showed limited utility, especially under warm and moist conditions as in our experiment (Konrad et al. 2021).

Other leaf traits that were examined did not show any sensitivity to $CO_2$ levels, including leaf surface pH, which was expected to decrease due to a higher dissolved inorganic carbon content in leaf surface solutions (Zeebe and Wolf-Gladrow 2001), but remained at 5.0 (Fig. 4). Variations among plant species in leaf surface pH range between 1.0 and 11.0, yet the pH of 5.0 measured here is lower than the pH values of other Rosids, typically between 6.0 and 10.0 (Gilbert and Renner 2021). In a preliminary experiment we observed black stains on some higher canopy leaves at 6000 ppm $CO_2$ (not shown), but these were probably related to high light conditions at the top of the mesocosm chamber. Finally, we did not detect any signs related to oxygen deficiency caused by the ultra-high $CO_2$ levels, as reported in humans at $CO_2 > 1,000$ ppm (Azuma et al. 2018).

Our experiment is not free of limitations. For example, a relatively low number of seedlings ($n = 10$) with natural interplant heterogeneity yielded high variation in some parameters (e.g., net assimilation; Fig. 3a). Our calculation method for net assimilation was also unique, since a cuvette gas analyzer could not be used at the high $CO_2$ levels studied here. A comparison between the methods, permitted at 400 ppm $CO_2$, indicated an underestimation by the mesocosm method (1.5 $\mu$mol $CO_2$ m$^{-2}$ s$^{-1}$ compared with 2.4 $\mu$mol $CO_2$ m$^{-2}$ s$^{-1}$ under PAR of $\mu$mol m$^{-2}$ s$^{-1}$ (Fig. S1). It is possible that, due to self-shading of the leaves, PAR levels were <100 $\mu$mol $CO_2$ m$^{-2}$ s$^{-1}$ for most leaves. Regardless, our calculation method was identical across the three $CO_2$ levels, and hence the relative changes persist. In addition, our choice of running the experiment over three consecutive periods (one for each of the three $CO_2$ concentrations), meant that plants were slightly different at each period. Such a difference was minimized by limiting the periods to just slightly over three weeks each, so that all treatments were applied on plants at the same developmental stage. As a result, the study results cannot be generalized to longer timescales of months and years. We reiterate that this choice of continuous experimentation with the

same plants eliminated potential effects of differences among plants and chambers, and that prior to each treatment, plants underwent a one-week acclimatization period within the chamber.

Studying the physiological responses to ultra-high $CO_2$ levels in evergreen tree species can inform our basic understanding of plant physiology as well as the cultivation of fruit trees under high anthropogenic $CO_2$ emission scenarios and in extreme environments, as on Mars (although not attempting to fully simulate Mars' conditions). Except for a few pioneering studies in small plants grown on artificial media in small chambers, it has been unknown whether plants can thrive or even survive under such high-$CO_2$ conditions, as has the effect of ultra-high $CO_2$ levels on plant water use. In this respect, our results bring encouraging prospects of three sorts: (1) *Guava* seedlings were actively growing at 6000 ppm $CO_2$, at a rate similar to their growth rate under ambient $CO_2$ levels. We note that other tree species may even show growth benefits, as observed at $CO_2$ between 400 and 1000 ppm (Paudel et al. 2018, Dror and Klein 2022, Norby et al. 2024). (2) Net assimilation increased significantly at 6000 ppm $CO_2$. This means that cultivation at elevated $CO_2$ can serve as a carbon sink, with a possible role in mitigation of anthropogenic $CO_2$ emissions. (3) Transpiration at 6000 ppm was 20% lower than at 400 ppm $CO_2$. Together with the increase in assimilation, we measured an almost four-fold increase in WUE across this $CO_2$ range. Our results are valuable for both future Earth (e.g., emission scenario RCP8.5) and Mars colonization (Richards et al. 2006, Wamelink et al. 2014, Slobodian et al. 2015). On Earth, indoor environments are already experiencing excessive $CO_2$ levels, and plants in green walls help to decrease indoor $CO_2$ levels (Agra et al. 2021). On Mars, low temperature and low atmospheric $O_2$ levels mean that plant cultivation will require an engineered environment. In this sense, our mesocosm can be regarded as a simulation of an artificial greenhouse (though we made no attempt to simulate Mars conditions). Considering the scarcity of water on Mars (Diez 2018, Nazari-Sharabian 2020) and in some regions on Earth, especially under climate change scenarios (Gosling et al. 2016, Rosa et al. 2020), the increase in WUE at high $CO_2$ levels observed in this study is a major advantage stemming from plant physiology. Overall, the high abundance of $CO_2$ in Earth's future atmosphere and in the present Martian atmosphere might compensate for water shortage.

**Acknowledgements**

The authors thank the Weizmann Center for Planetary Science for supporting the project.

**Authors' contributions**

IH and TK initiated the study. YBG and YBE established the experimental system and the first measurements. BEL performed the experiment and the analyses. BEL, TK, and IH wrote the manuscript.

**Competing interests**

The authors declare no conflict of interest in preparing this manuscript

**Open research**

Data will be available in an open repository upon manuscript acceptance

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

**Tables and Figures**

**Captions to Figures**

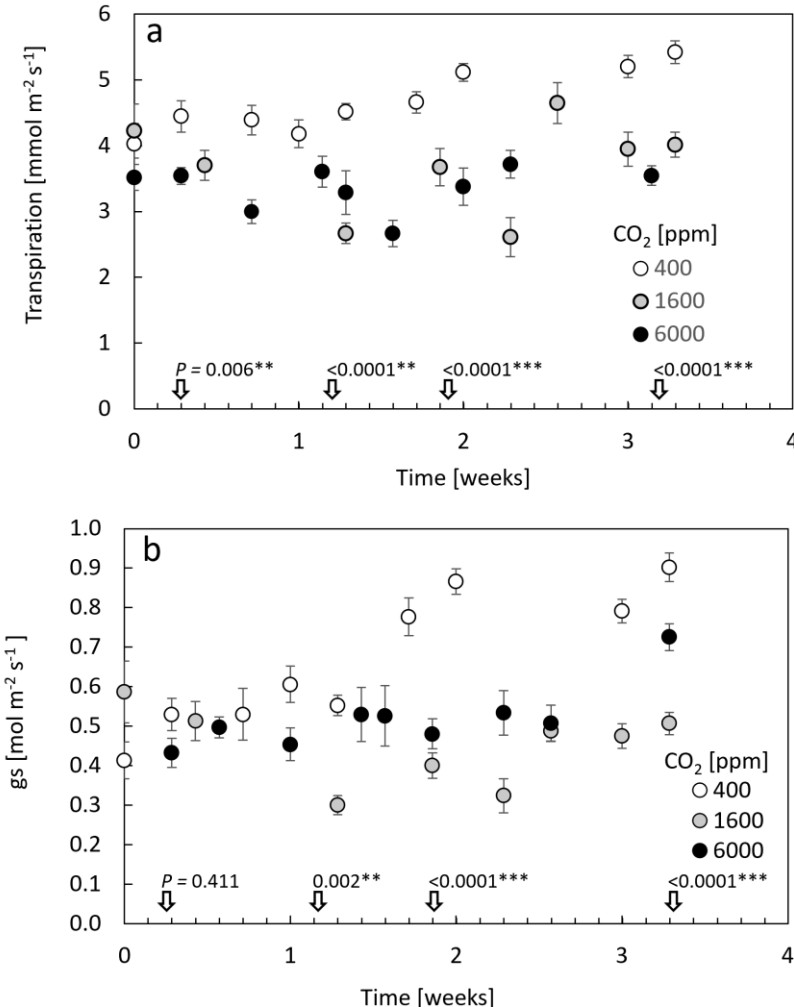


**Fig. 1. Leaf transpiration and stomatal conductance are decreased at 1600 ppm CO₂ and remain low at 6000 ppm.** Data points are means of 10 guava saplings subjected to different $CO_2$ concentrations. Measurements were made with a leaf cuvette. Error bars represent standard errors. *P*-values are from ANOVA on transpiration and stomatal conductance levels at specific dates (±1 day), and *, **, and *** indicate differences among $CO_2$ levels at 0.01, 0.001, and 0.0001 significance
levels.

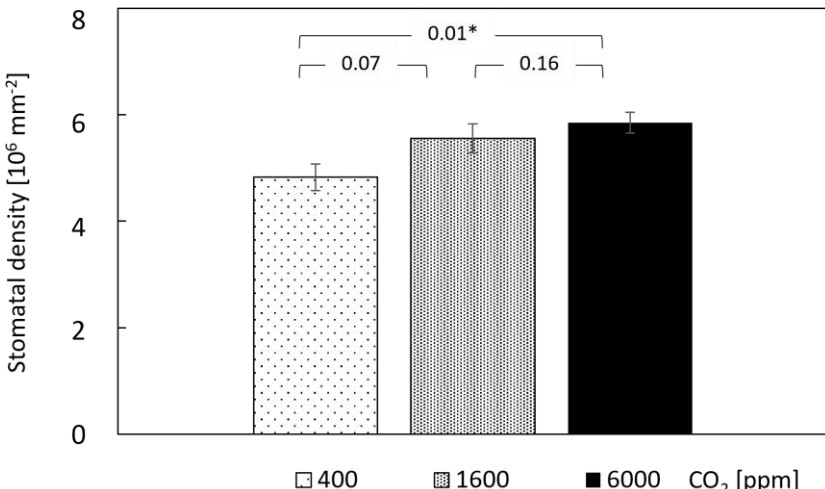

**Fig. 2. Stomatal density mildly increased under elevated CO₂.** Data points and bar heights are means of 10 guava saplings
subjected to different $CO_2$ concentrations. Error bars represent standard errors. *P*-values are from paired t-tests and *
indicates significant differences among $CO_2$ levels. Stomatal density was measured under a light microscope.

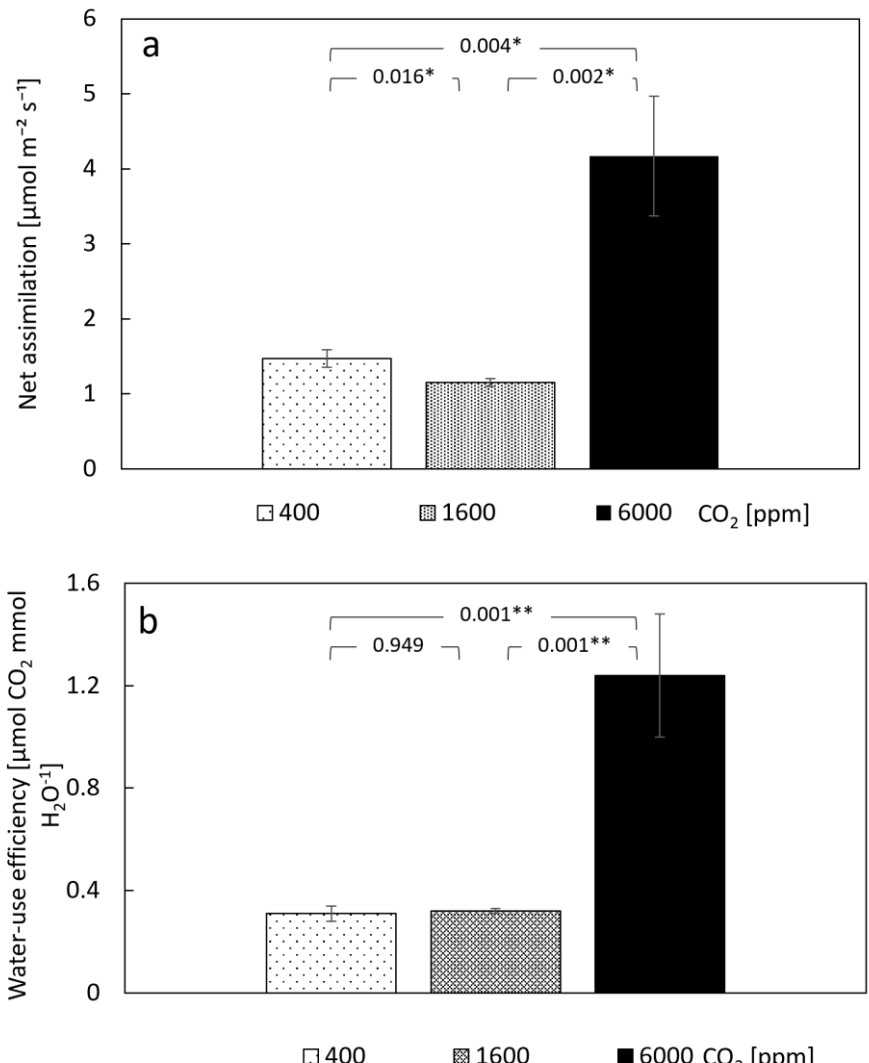

**Fig. 3. Net assimilation is mildly changed at 1600 ppm CO₂ and increases at 6000 ppm, driving a major increase in water-use efficiency.** Bar heights are means of 10 guava saplings subjected to different $CO_2$ concentrations. Error bars represent standard errors. *P*-values are from paired t-tests and * and ** indicate differences among $CO_2$ levels at 0.01 and 0.001 significance levels. Net assimilation (a) was calculated from chamber $CO_2$ dynamics. Water-use efficiency (b) was calculated as net assimilation divided by transpiration.

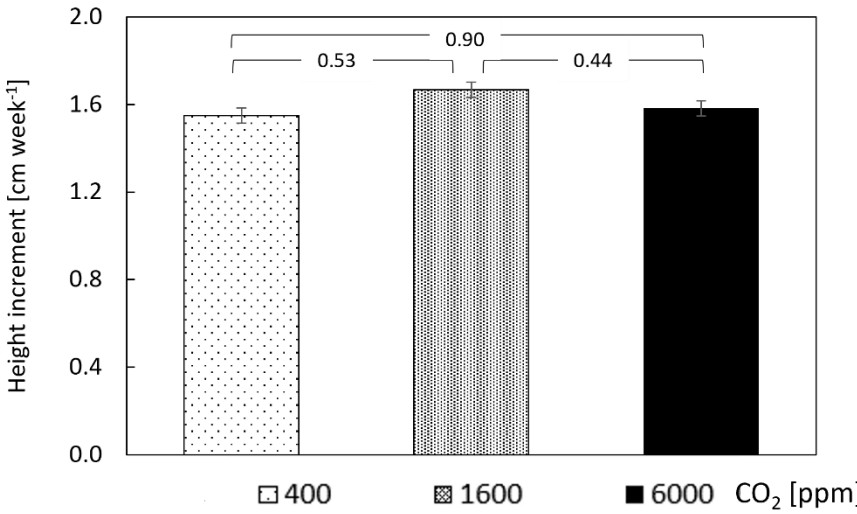


**Fig. 4. Plant growth is similar under different CO₂ concentrations**. Data points are means of 10 guava saplings subjected to different $CO_2$ concentrations. Error bars represent standard errors. *P*-values are from paired t-tests.