# Peer review of "Physiological responses to ultra-high CO2 levels in an evergreen tree species"

_EGUsphere, 2025_

## Author Response (AR1)

**Authors' responses to reviewers' comments on:**
**Physiological responses to ultra-high CO₂ levels in an evergreen tree species**

**Reviewer 1**
The manuscript by Levy et al. was a straightforward study showing the relatively short-term effect of 400, 1600 and 6000 ppm of CO2 on stomatal conductance and assimilation. The study exposed two-year old seedlings of guava trees to each CO2 level for a period of three weeks. They show a slight decrease in stomatal conductance and a somewhat increased assimilation with the 1600 and 6000ppm treatments. Overall the study only sufficiently showed a 3 week acclimation effect at differing CO2 levels. There were only two points of concern. First the, the measurement of stomatal density increasing at higher level of CO2. The manuscript does not make clear the types of leaves sampled for this measurement.

Response: We thank Reviewer 1 (Dr. Manandhar) for the valuable review of our manuscript, which helped improving it. Below please find our responses to each of the two concerns.

Which leaves were used to measure stomatal density to show the effect of CO2 treatment? Was there sufficient time for new leaves to develop and reach maturity during CO2 level exposure to determine whether there was a developmental effect on stomatal density?
Unless this study is suggesting that stomatal density is changing in the same mature leaf. If so, the manuscript needs provide evidence that the number of stomata on the same leaf increases or decreases after they reach maturity. Because at this moment there is no evidence of new regime of changes in cell fates in mature leaves. Papers on stomatal density signaling:
https://nph.onlinelibrary.wiley.com/doi/full/10.1111/j.1469-8137.2004.01292.x
https://pubmed.ncbi.nlm.nih.gov/16172139/
https://www.sciencedirect.com/science/article/pii/S1360138501020957

Response: Thank you for this comment. Information was added in the Methods section on leaf parameters: "Leaf stomatal density was determined on newly matured leaves formed in the new CO₂ environment (3 days old leaves from the top of the canopy, grown at PAR of 550 $\mu$mol m$^{-2}$ s$^{-1}$; $n$ = 10 per treatment) by calculating the number of stomata per unit area (mm$^2$)". In addition, the papers on stomatal density signaling were very relevant and interesting, and we now refer to two of them. Revisions were made in the Introduction: "Stomatal density has been hence used extensively as a paleo-CO₂ proxy (Steinthorsdottir et al. 2019, Konrad et al. 2021). However, this use has been rarely validated experimentally, but rather calibrated against other paleo-CO₂ proxies (Konrad et al. 2021; but see Brownlee 2001)". And in the Discussion: "Previous studies showed

that stomatal conductance of mature leaves has a regulatory effect on the stomatal development of expanding leaves (Brownlee 2001, Miyazawa et al. 2005). We do not fully understand why stomatal density increased, and whether this change represents a long-term response".

Second the authors could have verified or contrasted the result of photosynthesis induced stomatal closure saturating between 850 and 1600 could be verified with a gs-Ci curve and an Aci curve. Especially since the paper's objective is to show the gs and assimilation responses after acclimation for a few weeks. It would be valuable to compare with gs and A response to instantaneous changes in CO2 levels compared to three weeks of acclimation. Even an A-ci curve up to the maximum range that a li6800 can manage would have been enough to show this or the lack of li6800's ability to reach high enough CO2 levels to saturate A for this species.

Response: Point well taken. A new figure (Fig. S3) was added to the manuscript, showing the results of this pilot experiment as suggested by the reviewer. Consequently, text was added in the Methods, in the Results, and in the Discussion.
In the Methods: "A second pilot experiment tested the plant response to instantaneous changes in $CO_2$ level. Here, three plants grown at ambient $CO_2$ level were exposed to increasing $CO_2$ levels and their net assimilation rate ($\mu$mol $CO_2$ m$^{-2}$ s$^{-1}$) and stomatal conductance (mol m$^{-2}$ s$^{-1}$) were measured with the LI-6800 infrared gas analyzer. Temperature, relative humidity, and PAR level within the leaf cuvette were set to 26 ºC, 75%, and 600 $\mu$mol m$^{-2}$ s$^{-1}$, respectively. $CO_2$ level was adjusted to 400, 100, 50, 0, 150, 250, 300, 400, 500, 600, 700, 800, 900, 1000, 1100, 1200, 1300, 1400, 1500, 1600, 1700, 1800, 1900, 2000 and 400 $\mu$mol $CO_2$ m$^{-2}$ s$^{-1}$, in this order".
In the Results: "Plants that were grown at ambient $CO_2$ level were also tested for their A and gs responses to instantaneous changes in $CO_2$ level (Fig. S3). A increased linearly from -2.3±0.6 $\mu$mol $CO_2$ m$^{-2}$ s$^{-1}$ at 0 ppm $CO_2$ to 12.9±1.0 $\mu$mol $CO_2$ m$^{-2}$ s$^{-1}$ at 800 ppm $CO_2$ (with a compensation point around 100 ppm $CO_2$). At higher $CO_2$ levels, A nearly saturated, yet continued to increase up to 15.7±0.8 $\mu$mol $CO_2$ m$^{-2}$ s$^{-1}$ at 2000 ppm $CO_2$, the highest level permitted by the IRGA instrument. In parallel, gs decreased in two measured plants from 0.14 to 0.11 mol m$^{-2}$ s$^{-1}$, and from 0.09 to 0.05 mol m$^{-2}$ s$^{-1}$.".
And in the Discussion: "Integrating our current and previous experiments, we can conclude that for an evergreen fruit tree such as guava, $CO_2$-induced stomatal closure probably saturates between 850 and 1600 ppm $CO_2$. Do these responses reflect an adaptation to elevated $CO_2$, or an instantaneous response? Experimenting with trees with no previous exposure to elevated $CO_2$, we observed a similar trend of stomatal closure (a 20-50% reduction in gs between 400 to 1600 ppm $CO_2$; Fig. S3) which seemed to level off at higher $CO_2$ levels. Despite this reduction, net carbon assimilation increased, mostly below 800 ppm $CO_2$, while measurements were limited to 2000 ppm $CO_2$, due to the IRGA constraints".

**Reviewer 2**

The authors exposed two-year old dwarf guava seedlings (Psidium cattleyanum) to a range of CO2 concentrations, up to a very high level of 6000 ppm, evaluating the impact on transpiration, assimilation and water use efficiency. Overall, the question is interesting and posed in an engaging context of larger questions. However, many conclusions are stated as if they apply to all plants or all trees, where we actually have research here on a single species, which happens to be a small stature tree or shrub, rather than a typical canopy tree of the type in which much of the aboveground biomass is stored in tropical rainforests. This makes many conclusions a bit overstated.

Response: We thank Reviewer 2 for the valuable comments and insights which helped improving our manuscript. We acknowledge that our study was on a single tree species and does not represent all trees or all plants. We did our best to avoid overstatements, e.g., in the abstract and discussion: "Taken together, our measurements demonstrated both the potential and limit of $CO_2$-induced stomatal closure, with positive implications for fruit tree growth in ultra-high $CO_2$ environments...". The concluding paragraph is also specific to the species under study: "In this respect, our results bring encouraging prospects of three sorts: (1) *Guava* seedlings were actively growing at 6000 ppm $CO_2$, at a rate similar to their growth rate under ambient $CO_2$ levels. We note that other tree species may even show growth benefits, as observed at $CO_2$ between 400 and 1000 ppm (Paudel et al. 2018, Dror and Klein 2022)".

The research here is on an economically important plant in some regions where it is cultivated, as well as some (like Hawai'i) where it is a nuisance or invasive species. The cultivation context might be more interesting, especially for the exobiological framing of the research at very high concentrations.

Response: We thank the reviewer for these insights. Based on these, information was added in the Introduction, along with two new references: "This short-stature tree species (growing up to 6 m in height) is also well-adapted to medium light levels. Dwarf guava is an economically important species in some areas where it is cultivated (Patel 2012) and an invasive species in others, like Hawaii (Huenneke et al. 1990)".

I have some concerns on the possible errors introduced by calculating assimilation by using CO2 sensors meant for chamber regulation rather than...

Response: Please see in the point-by-point responses below.

Finally, a greater number and diversity of citations are needed to support many points made here, with less dependence on previous work involving the senior author.

Response: Point well taken. Based on the comments made here and those made by the other reviewer, nine new citations were added to the manuscript, none related to the former studies of the authors.

I concur with the previous comments on stomatal density and possibly using the LI-1680 to confirm results, which should be/would have been possible if impacts were observed below 2000 ppm.

Response: Point well taken. A new figure (Fig. S3) was added to the manuscript, showing the results of a pilot experiment as suggested by the reviewer. Consequently, text was added in the Methods, in the Results, and in the Discussion.
In the Methods: "A second pilot experiment tested the plant response to instantaneous changes in $CO_2$ level. Here, three plants grown at ambient $CO_2$ level were exposed to increasing $CO_2$ levels and their net assimilation rate ($\mu$mol $CO_2$ m$^{-2}$ s$^{-1}$) and stomatal conductance (mol m$^{-2}$ s$^{-1}$) were measured with the LI-6800 infrared gas analyzer. Temperature, relative humidity, and PAR level within the leaf cuvette were set to 26 $^{\circ}$C, 75%, and 600 $\mu$mol m$^{-2}$ s$^{-1}$, respectively. $CO_2$ level was adjusted to 400, 100, 50, 0, 150, 250, 300, 400, 500, 600, 700, 800, 900, 1000, 1100, 1200, 1300, 1400, 1500, 1600, 1700, 1800, 1900, 2000 and 400 $\mu$mol $CO_2$ m$^{-2}$ s$^{-1}$, in this order".
In the Results: "Plants that were grown at ambient $CO_2$ level were also tested for their A and gs responses to instantaneous changes in $CO_2$ level (Fig. S3). A increased linearly from -2.3±0.6 $\mu$mol $CO_2$ m$^{-2}$ s$^{-1}$ at 0 ppm $CO_2$ to 12.9±1.0 $\mu$mol $CO_2$ m$^{-2}$ s$^{-1}$ at 800 ppm $CO_2$ (with a compensation point around 100 ppm $CO_2$). At higher $CO_2$ levels, A nearly saturated, yet continued to increase up to 15.7±0.8 $\mu$mol $CO_2$ m$^{-2}$ s$^{-1}$ at 2000 ppm $CO_2$, the highest level permitted by the IRGA instrument. In parallel, gs decreased in two measured plants from 0.14 to 0.11 mol m$^{-2}$ s$^{-1}$, and from 0.09 to 0.05 mol m$^{-2}$ s$^{-1}$.".
And in the Discussion: "Integrating our current and previous experiments, we can conclude that for an evergreen fruit tree such as guava, $CO_2$-induced stomatal closure probably saturates between 850 and 1600 ppm $CO_2$. Do these responses reflect an adaptation to elevated $CO_2$, or an instantaneous response? Experimenting with trees with no previous exposure to elevated $CO_2$, we observed a similar trend of stomatal closure (a 20-50% reduction in gs between 400 to 1600 ppm $CO_2$; Fig. S3) which seemed to level off at higher $CO_2$ levels. Despite this reduction, net carbon assimilation increased, mostly below 800 ppm $CO_2$, while measurements were limited to 2000 ppm $CO_2$, due to the IRGA constraints".

Lines 158-159: "Notably, at 6000 ppm CO2, fluctuations of ±30 ppm are equal to ±0.5%, which is negligible." This would be a valid statement if the whole experiment was

conducted at this level, but it is a much larger percent deviation at 400 ppm.  More importantly, however, is that this sensor is used to estimate the assimilation at the elevated CO2 levels by all the plants in the chamber.  So the relevant percent error is that relative to the difference in CO2 concentrations being used to estimate assimilation, not the absolute CO2 level in the chamber.

Response: Thank you for this important comment. The sentence was revised accordingly: "Notably, at 400, 1600, and 6000 ppm $CO_2$, fluctuations of ±30 ppm are equal to ±7.5%, 1.9%, and 0.5%, which are small". We understand the concern of the reviewer about the measurement accuracy of our sensor and its implications. To our understanding, using the rate of decrease in the $CO_2$ concentration (ppm) within the controlled mesocosm chamber over a known duration should be more sensitive to sensor's precision than accuracy. In other words, since the same sensor was used throughout the experiment, and the difference between two time-points was used for each value, the sensor's precision (how close were measurements of the same $CO_2$ concentration) was more important than its accuracy (how close were measurements to the true $CO_2$ concentration). In terms of precision, we found the SCD30 sensor highly reliable. Evidencing that were the smooth $CO_2$ concentration curves observed for each of the 20 examination periods used for assimilation calculation at each $CO_2$ level.

Lines 194-195: "using the LI-6800 infrared gas analyzer (see under Plant response to light intensity) but could not be measured using the leaf cuvette at the higher CO2 levels, due to the instrument limitations".  I believe the LI-6800 can hold concentrations up to 2000 umol mol-1, so measurements at the 1600 ppm level should have been achievable.  Additionally, an A-Ci curve would likely show flattening at a concentration well below 2000 ppm, even in plants growing in 6000 ppm.  These could have been used to confirm the results from the whole chamber estimates of assimilation.

Response: Point well taken. Please see the details in the response to the comment before the previous.

Lines 104-105: "it has been established that under field conditions, tree growth is unaffected by elevated CO2 (Korner et al. 2005, Klein et al. 2016)"  This statement is misleading and relies on results from a single site.  This statement is not supported by some free air carbon dioxide (FACE) studies, despite the results at the elevated CO2 facility in the Swiss Alps cited here.   While similar results have been found in some FACE studies (e.g. Norby, R.J., et al. Tree Physiology 42.3 (2022): 428-440 or Jiang, M.K. et al. Nature 580, 227–231 (2020), authors should note that these or other citations could bolster the argument here), other FACE studies have found increased growth rates (e.g. Kim, Dohyoung, et al. Global Change Biology 26.4 (2020): 2519-2533 or  Norby, Richard J., et al. Nature Climate Change 14.9 (2024): 983-988.)  Overall, the

preponderance of evidence is that individual trees will grow faster under elevated CO2 unless they run into another growth limitation. The difference is not merely if the trees are under 'field conditions', but depends on growth limits imposed by factors such as nutrient supply or light availability (e.g., open or closed canopy). These limitations are uncommon and usually remedied in cultivated fruit-bearing species, so arguably do not apply to this research on guava trees. The lack of response is generally associated with forests that have closed canopies and applies to growth at the stand scale. It is not generally accepted to apply to individual trees in an open canopy unless another severe growth limitation is in place.

Response: Thank you for this point. The text has been revised to provide the broader context, also citing these key citations: "The higher assimilation can lead to increased carbon storage (Kinsman et al. 1997, Paudel et al. 2018) and increased growth. For example, elevated $CO_2$ levels resulted in elongation of branches and stems by 33% in *Garcinia mangostana* (Downton et al. 1990) and by 15% in *Glycine max* (Rogers et al. 1992). However, in contrast to growth in controlled environments, it has been established that under field conditions, where competition over limited resources prevails, tree growth is mostly unaffected by elevated $CO_2$ (Korner et al. 2005, Klein et al. 2016, Jiang et al. 2020, Norby et al. 2022). Yet, increased tree growth is still observed in some cases (Kim et al. 2020, Norby et al. 2024)". Text was also revised in the concluding paragraph of the Discussion: "*Guava* seedlings were actively growing at 6000 ppm $CO_2$, at a rate similar to their growth rate under ambient $CO_2$ levels. We note that other tree species may even show growth benefits, as observed at $CO_2$ between 400 and 1000 ppm (Paudel et al. 2018, Dror and Klein 2022, Norby et al. 2024)"

Lines 115-116: "High concentrations of CO2, which decrease the stomatal conductance, lead to a notable reduction in water loss and an increase in the plant's water-use efficiency (WUE)…" This statement is not fully supported by the literature. A meta analysis of intrinsic WUE found that increased photosynthesis, rather than decreased stomatal conductance was primarily responsible for increased iWUE under elevated CO2 (Mathias, J.M., and R.B. Thomas. Proceedings of the National Academy of Sciences 118.7 (2021): e2014286118.), while results from some sites show that results at the stand scale may depend on whether increases in canopy leaf area offset stomatal closure (Ward, E.J., et al. Global Change Biology 24.10 (2018): 4841-4856), which is more likely with well-spaced, managed fruit and fiber plantation forests than with natural, mature multispecies stands.

Response: Thank you for these interesting points, which add context and perspective to our manuscript. Text has been revised in the Introduction: "High concentrations of $CO_2$, which increase $CO_2$ assimilation and decrease the stomatal conductance, lead to a notable reduction in water loss and an increase in the plant's water-use efficiency

(WUE), the ratio of net $CO_2$ assimilation to transpiration, representing the plant's ability to maximize carbon gain while minimizing water loss (Field et al. 1995, Mathias and Thomas 2021, Dror and Klein 2022)".

**Reviewer 3**

This manuscript presents experimental results on leaf gas exchange and carbon assimilation under ambient and extremely high CO2 concentrations. In the experiment, fruit trees were exposed to $CO_2$ levels of 400, 1600, and 6000 ppm (approximating Mars' atmospheric conditions) in a controlled mesocosm. Leaf transpiration, stomatal conductance, stomatal density, CO2 assimilation, and growth metrics were systematically monitored. Overall, the experiment is robust, and the results are generally well interpreted. I have a few outstanding questions, particularly regarding the interpretation of WUE changes, statistical analysis, and stomatal density. I also note a few methodological aspects that need additional details for clarity.

**1) Analysis and discussion of the unchanged WUE from 400 to 1600ppm $CO_2$**

The manuscript reports no increase in WUE from 400 to 1600 ppm CO2, attributed to a decrease in assimilation due to stronger stomatal conductance reduction not compensated by passive diffusion. Since many stomatal conductance models predict a near-linear increase of WUE to Ca, with no indication of saturation (e.g. Walker et al., 2021), further analysis and discussion would be helpful to explain the observed stable WUE considering other limitations (e.g. mesophyll conductance) as well as the potential uncertainties/biases in the measurements (also see point 4).

Walker, A. P., De Kauwe, M. G., Bastos, A.et al..: Integrating the evidence for a terrestrial carbon sink caused by increasing atmospheric CO2, New Phytologist, 229, 2413–2445, https://doi.org/10.1111/nph.16866, 2021.

Response: Thank you for these interesting points, which add context and perspective to our manuscript. Text has been revised in the Introduction: "High concentrations of $CO_2$, which increase $CO_2$ assimilation and decrease the stomatal conductance, lead to a notable reduction in water loss and an increase in the plant's water-use efficiency (WUE), the ratio of net $CO_2$ assimilation to transpiration, representing the plant's ability to maximize carbon gain while minimizing water loss (Field et al. 1995, Mathias and Thomas 2021, Dror and Klein 2022)". In addition, text has been added in the Discussion: "Noteworthy, the stable WUE (400-1600 ppm $CO_2$) was driven by proportional reductions in assimilation and transpiration, whereas the major WUE increase (at the 1600-6000 ppm $CO_2$ transition) was driven by increased assimilation. The stable WUE (400-1600 ppm $CO_2$) contrasts an overall increase of WUE in the terrestrial biosphere at this $CO_2$ range (Walker et al. 2021), however can be expected due to the reduced

stomatal conductance in a broadleaf tropical tree species (Paudel et al. 2018), which is not a ubiquitous response across tree species (Klein and Ramon 2019)".

Regarding our net assimilation measurements, we apologize for not being sufficiently clear in the text, which has been revised now: "The rate of $CO_2$ assimilation was measured with a leaf cuvette at 400 ppm µmol $CO_2$ using the LI-6800 infrared gas analyzer (see under *Plant response to light intensity*) but could not be measured using the leaf cuvette at the higher $CO_2$ levels, due to the instrument limitations. Instead, assimilation rate was measured across the three $CO_2$ levels by analyzing the rate of decrease in the $CO_2$ concentration (ppm) within the controlled mesocosm chamber over a known duration (measured in s), which was then divided by the corresponding leaf surface area of all plants ($m^2$)".

We now also discuss the comparison between the methods in the Discussion paragraph describing the limitations of the study "Our calculation method for net assimilation was also unique, since a cuvette gas analyzer could not be used at the high $CO_2$ levels studied here. A comparison between the methods, permitted at 400 ppm $CO_2$, indicated an underestimation by the mesocosm method (1.5 µmol $CO_2$ $m^{-2}$ $s^{-1}$ compared with 2.4 µmol $CO_2$ $m^{-2}$ $s^{-1}$ under PAR of µmol $CO_2$ $m^{-2}$ $s^{-1}$ (Fig. S1). It is possible that, due to self-shading of the leaves, PAR levels were <100 µmol $CO_2$ $m^{-2}$ $s^{-1}$ for most leaves. Regardless, our calculation method was identical across the three $CO_2$ levels, and hence the relative changes persist".

**2) Statistical analysis of leaf transpiration and stomatal conductance**

2a) A summary statistical test aggregating data across all days for each CO2 level would be helpful for interpretation. While the current presentation is informative incorporating all data, it is somewhat difficult to interpret.

2b) Moreover, it is unclear how the reported 20% differences in transpiration (Line 231) and in 50% decrease in gs (Line 234) between CO2 levels was derived, and how statistically significant they are. Further clarification would be helpful.

2c) The observed increasing trend of gs (pretty significant) and transpiration under 400 ppm is attributed to acclimation within the first 10 days. However, both variables seem to continue increasing beyond 10 days. Could the authors provide potential explanation for this?

Response: Per the reviewer's comment, information was added to the text: "Examining leaf transpiration (Fig. 1a), a 20% decrease was observed upon the transition from 400 to 1600 ppm $CO_2$ (from ~4.4 to ~3.5 mmol $m^{-2}$ $s^{-1}$; F=44.9, *P*<0.0001). The subsequent increase in $CO_2$ concentration to 6000 ppm resulted in no statistically significant changes in leaf transpiration compared to the 1600 ppm treatment, suggesting a potential threshold effect in the modulation of transpiration rates by elevated $CO_2$. In terms of stomatal conductance (gs), the transition from 400 to 1600 ppm $CO_2$ resulted in a 50% reduction in gs, which persisted at the 6000 ppm treatment (from ~0.85 to

~0.42 mol m$^{-2}$ s$^{-1}$; F=22.6, *P*<0.0001; Fig. 1b)". Regarding the increasing trends of gs and T under 400 ppm, we note that these increases occurred within the 10 days indicated in the text, and maintained later.

**3) Analysis of leaf stomatal density does not account for potential confounding factors**

3a) In terms of measurement (L210), more details would be needed regarding the sampling frequency, number of leaf samples each time, leaf age, leaf location – top of or below canopy.

3b) As noted by other reviewers, the analysis of leaf stomatal density does not appear to account for confounding factors, such as leaf age, leaf light environment, specific trees. Stomatal density change might only occur in new leaves during development, probably wouldn't happen in mature leaves.

3c) Regarding Fig. 1:

- It's not always clear, which day and which measurements each p-value correspond to.

- In abstract and methodology, the treatment periods were three weeks, but this figure appears to cover four weeks for each treatment.

Response: Thank you for this comment. Information was added in the Methods section on leaf parameters: "Leaf stomatal density was determined on newly matured leaves formed in the new $CO_2$ environment (3 days old leaves from the top of the canopy, grown at PAR of 550 µmol m$^{-2}$ s$^{-1}$; *n* = 10 per treatment) by calculating the number of stomata per unit area (mm$^2$)". In addition, revisions were made in the Introduction: "Stomatal density has been hence used extensively as a paleo-$CO_2$ proxy (Steinthorsdottir et al. 2019, Konrad et al. 2021). However, this use has been rarely validated experimentally, but rather calibrated against other paleo-$CO_2$ proxies (Konrad et al. 2021; but see Brownlee 2001)". And in the Discussion: "Previous studies showed that stomatal conductance of mature leaves has a regulatory effect on the stomatal development of expanding leaves (Brownlee 2001, Miyazawa et al. 2005). We do not fully understand why stomatal density increased, and whether this change represents a long-term response".

Regarding Fig. 1, the caption was adjusted for clarification: "*P*-values are from ANOVA on transpiration and stomatal conductance levels at specific dates, and *, **, and *** indicate differences among $CO_2$ levels at 0.01, 0.001, and 0.0001 significance levels".

Regarding the span of each treatment, the text was corrected in the abstract: "Plant growth, and leaf gas exchange (transpiration, stomatal conductance, and $CO_2$ assimilation) were measured on a weekly basis for 23-25 consecutive days"; in the methodology: "To eliminate potential effects of differences among plants and chambers, the entire experiment was conducted on the same plants within the same chamber. Therefore, treatment periods were limited to just over three weeks (23-25 days), so that all treatments were applied on plants at the same developmental stage";

and in the discussion: "Such a difference was minimized by limiting the periods to just slightly over three weeks each, so that all treatments were applied on plants at the same developmental stage".

**4) Measurement and analysis of net assimilation**
4a) Net assimilation was measured with a gas analyzer under 400ppm, and inferred from chamber CO2 concentration under 1600 and 6000ppm (L193). Were the two approaches cross-compared under 400ppm levels? This would be helpful to verify consistency and understand potential biases.
4b) The reported decrease in assimilation from 400 to 1600 is interesting. It would be helpful to rule out any potential impacts from the change of measurement method.

Response: Thank you for this comment. We apologize for not being sufficiently clear in the text, which has been revised now: "The rate of $CO_2$ assimilation was measured with a leaf cuvette at 400 ppm µmol $CO_2$ using the LI-6800 infrared gas analyzer (see under *Plant response to light intensity*) but could not be measured using the leaf cuvette at the higher $CO_2$ levels, due to the instrument limitations. Instead, assimilation rate was measured across the three $CO_2$ levels by analyzing the rate of decrease in the $CO_2$ concentration (ppm) within the controlled mesocosm chamber over a known duration (measured in s), which was then divided by the corresponding leaf surface area of all plants ($m^2$)".
We now also discuss the comparison between the methods in the Discussion paragraph describing the limitations of the study "Our calculation method for net assimilation was also unique, since a cuvette gas analyzer could not be used at the high $CO_2$ levels studied here. A comparison between the methods, permitted at 400 ppm $CO_2$, indicated an underestimation by the mesocosm method (1.5 µmol $CO_2$ $m^{-2}$ $s^{-1}$ compared with 2.4 µmol $CO_2$ $m^{-2}$ $s^{-1}$ under PAR of µmol $CO_2$ $m^{-2}$ $s^{-1}$ (Fig. S1). It is possible that, due to self-shading of the leaves, PAR levels were <100 µmol $CO_2$ $m^{-2}$ $s^{-1}$ for most leaves. Regardless, our calculation method was identical across the three $CO_2$ levels, and hence the relative changes persist".
Please also note that a new figure (Fig. S3) was added to the manuscript, showing the results of this pilot experiment showing that leaf assimilation increases and stomatal conductance decreases under instantaneous exposure to elevated $CO_2$ concentrations. Consequently, text was added in the Methods, in the Results, and in the Discussion.
In the Methods: "A second pilot experiment tested the plant response to instantaneous changes in $CO_2$ level. Here, three plants grown at ambient $CO_2$ level were exposed to increasing $CO_2$ levels and their net assimilation rate (µmol $CO_2$ $m^{-2}$ $s^{-1}$) and stomatal conductance (mol $m^{-2}$ $s^{-1}$) were measured with the LI-6800 infrared gas analyzer. Temperature, relative humidity, and PAR level within the leaf cuvette were set to 26 ºC, 75%, and 600 µmol $m^{-2}$ $s^{-1}$, respectively. $CO_2$ level was adjusted to 400, 100, 50, 0, 150,

250, 300, 400, 500, 600, 700, 800, 900, 1000, 1100, 1200, 1300, 1400, 1500, 1600, 1700, 1800, 1900, 2000 and 400 µmol $CO_2$ $m^{-2}$ $s^{-1}$, in this order".

In the Results: "Plants that were grown at ambient $CO_2$ level were also tested for their A and gs responses to instantaneous changes in $CO_2$ level (Fig. S3). A increased linearly from -2.3±0.6 µmol $CO_2$ $m^{-2}$ $s^{-1}$ at 0 ppm $CO_2$ to 12.9±1.0 µmol $CO_2$ $m^{-2}$ $s^{-1}$ at 800 ppm $CO_2$ (with a compensation point around 100 ppm $CO_2$). At higher $CO_2$ levels, A nearly saturated, yet continued to increase up to 15.7±0.8 µmol $CO_2$ $m^{-2}$ $s^{-1}$ at 2000 ppm $CO_2$, the highest level permitted by the IRGA instrument. In parallel, gs decreased in two measured plants from 0.14 to 0.11 mol $m^{-2}$ $s^{-1}$, and from 0.09 to 0.05 mol $m^{-2}$ $s^{-1}$.".

And in the Discussion: "Integrating our current and previous experiments, we can conclude that for an evergreen fruit tree such as guava, $CO_2$-induced stomatal closure probably saturates between 850 and 1600 ppm $CO_2$. Do these responses reflect an adaptation to elevated $CO_2$, or an instantaneous response? Experimenting with trees with no previous exposure to elevated $CO_2$, we observed a similar trend of stomatal closure (a 20-50% reduction in gs between 400 to 1600 ppm $CO_2$; Fig. S3) which seemed to level off at higher $CO_2$ levels. Despite this reduction, net carbon assimilation increased, mostly below 800 ppm $CO_2$, while measurements were limited to 2000 ppm $CO_2$, due to the IRGA constraints".

**5) How did leaf surface areas change across treatments?**
L295: How about leaf area data measured but not shown in the manuscript?

Response: Point well taken. Per the reviewer's comment, a new figure (Fig. S2) was added in the supplementary information. The text was changed accordingly: "Measurements of the weekly growth height did not reveal a significant change (Fig. 4), nor was the weekly increment of leaf surface area (Fig. S2)".

---

## Author Response (AR2)

**Authors' responses to reviewers' comments on:**
**Physiological responses to ultra-high CO$_2$ levels in an evergreen tree species**

**Reviewer 1**
I appreciate the authors' efforts in addressing my previous comments and for the comprehensive revision, which has significantly improved the clarity and overall quality of the manuscript. The writing is clear, and the experimental results are thoughtfully presented and discussed. I only have a few remaining suggestions and questions regarding clarity:

Response: We thank Reviewer 1 (Dr. Manandhar) for the valuable review of our manuscript, which helped improving it. Below please find our responses to each of the remaining issues.

L31: The sentence "In summary, ultra-high CO2 may partly compensate for water shortage." This summary sentence is a bit broad and lacks content. Consider rephrasing "water shortage" – a broad term with terms specific to plant physiology, such as "stress" or "limited water availability".

Response: The sentence was revised per the reviewer's suggestion: "In summary, ultra-high CO2 may partly compensate for limited water availability".

L35: Consider rephrasing "the very fabric of life", as it feels a bit rhetorical.

Response: Correct. The sentence was revised per the reviewer's suggestion: "Understanding the implications of elevated atmospheric CO$_2$ levels on plant physiology is paramount, as it has profound consequences for global ecosystems, water economy, and terrestrial carbon cycling (Bazzaz et al. 1990)".

L37: "a response is triggered" this is somewhat broad and redundant

Response: The sentence was revised per the reviewer's suggestion: "When the level of CO$_2$ within the plant diminishes, a regulatory response is triggered within the stomatal guard cells, and the stomata open, allowing diffusive uptake of atmospheric CO$_2$ (Lawson and Morison 2004)".

L50: "adequate nutrient availability": "adequate" seems redundant

Response: The sentence was revised: "Fluctuations in nutrient availability, particularly potassium, can influence stomatal conductance by affecting the turgor pressure of guard cells, which control stomatal opening and closing (Lebaudy et al. 2008)".

L157, 159: It's not clear if one week is sufficient for plant acclimation though; Poor et al. 2020 doesn't provide an assessment of acclimation. Also, Poorter et al. 2020 is not currently in the Reference list.

Response: We thank the reviewer for catching the missing reference, which was now added. The text is now clarified: "...treatment periods were limited to just over three weeks (23-25 days), so that all treatments were applied on plants at the same developmental stage. This period is sufficient to allow for plant acclimation to the $CO_2$ treatment (Poorter et al. 2022). The experiment was conducted between 4 June and 27 August 2023. Plants were first exposed to 400 ppm $CO_2$, followed by 1600 ppm, and then 6000 ppm, i.e., from ambient to high and ultra-high $CO_2$. Prior to each treatment, plants underwent an additional one-week acclimation period within the mesocosm chamber".

L183: How long did each CO2 level last?

Response: The information was added to the text: "$CO_2$ level was adjusted every 3-4 minutes to 400, 100, 50, 0, 150, 250, 300, 400, 500, 600, 700, 800, 900, 1000, 1100, 1200, 1300, 1400, 1500, 1600, 1700, 1800, 1900, 2000 and 400 µmol $CO_2$ $m^{-2}$ $s^{-1}$, in this order".

L220: "n=10 per treatment" could be further clarified. It's unclear what does "n" denote.

Response: We thank the reviewer for catching this error. Text was changed to: "n=10 plants".

Fig. 1: A couple of questions regarding clarity:
- The colors of 1600ppm and 6000ppm are difficult to distinguish.
- The x-axis minor ticks are not visible, but seems to correspond to individual days. Clarifying this in axis label would be helpful.
- It remains difficult to interpret which data points the P-values refer to. For example, the first p-value (P=0.411) seems to compare one 400ppm and one 6000ppm measurement, and it's unclear if the adjacent 1600 measurement was also included.

Response: Points well taken. The 1600 ppm data points were changed to a lighter shade of grey. X-axis minor tick marks were added for clarity, and the caption was revised for clarity: "**Fig. 1. Leaf transpiration and stomatal conductance are decreased at 1600 ppm $CO_2$ and remain low at 6000 ppm.** Data points are means of 10 guava saplings subjected to different $CO_2$ concentrations. Measurements were made with a leaf cuvette. Error bars represent standard errors. *P*-values are from ANOVA on transpiration

and stomatal conductance levels at specific dates (±1 day), and *, **, and *** indicate differences among $CO_2$ levels at 0.01, 0.001, and 0.0001 significance levels".

L364: Potential typo "under PAR of µmol CO2 $m^{-2}$ $s^{-1}$"

Response: We thank the reviewer for catching this error. Text was changed to: "under PAR of µmol $m^{-2}$ $s^{-1}$".